# Artificial selection methods from evolutionary computing show promise for directed evolution of microbes

Alexander Lalejini[1,2]*, Emily Dolson[3,4], Anya E Vostinar[5], Luis Zaman[1,2]*

[1]Department of Ecology and Evolutionary Biology, University of Michigan, Ann Arbor, United States; [2]Center for the Study of Complex Systems, University of Michigan, Ann Arbor, United States; [3]Department of Computer Science and Engineering, Michigan State University, East Lansing, United States; [4]Program in Ecology, Evolution, and Behavior, Michigan State University, East Lansing, United States; [5]Computer Science Department, Carleton College, Northfield, United States

**Abstract** Directed microbial evolution harnesses evolutionary processes in the laboratory to construct microorganisms with enhanced or novel functional traits. Attempting to direct evolutionary processes for applied goals is fundamental to evolutionary computation, which harnesses the principles of Darwinian evolution as a general-purpose search engine for solutions to challenging computational problems. Despite their overlapping approaches, artificial selection methods from evolutionary computing are not commonly applied to living systems in the laboratory. In this work, we ask whether parent selection algorithms—procedures for choosing promising progenitors—from evolutionary computation might be useful for directing the evolution of microbial populations when selecting for multiple functional traits. To do so, we introduce an agent-based model of directed microbial evolution, which we used to evaluate how well three selection algorithms from evolutionary computing (tournament selection, lexicase selection, and non-dominated elite selection) performed relative to methods commonly used in the laboratory (elite and top 10% selection). We found that multiobjective selection techniques from evolutionary computing (lexicase and non-dominated elite) generally outperformed the commonly used directed evolution approaches when selecting for multiple traits of interest. Our results motivate ongoing work transferring these multi-objective selection procedures into the laboratory and a continued evaluation of more sophisticated artificial selection methods.

*For correspondence:
lalejini@umich.edu (AL);
zamanlh@umich.edu (LZ)

**Competing interest:** The authors declare that no competing interests exist.

## Editor's evaluation

The study offers a valuable contribution to the field. While the fields of artificial life and experimental evolution in microbes have been connected for many years, there have been few studies to meaningfully demonstrate how work in evolutionary computation can meaningfully inform the design and execution of microbial experiments. This study represents a truly innovative approach and may fuel further studies at the intersection between computational evolution and experimental evolution.

## Introduction

Directed evolution harnesses laboratory artificial selection to generate biomolecules or organisms with desirable functional traits (*Arnold, 1998*; *Sánchez et al., 2021*). The scale and specificity of artificial selection has been revolutionized by a deeper understanding of evolutionary and molecular biology in combination with technological innovations in sequencing, data processing, laboratory

**eLife digest** Humans have long known how to co-opt evolutionary processes for their own benefit. Carefully choosing which individuals to breed so that beneficial traits would take hold, they have domesticated dogs, wheat, cows and many other species to fulfil their needs.

Biologists have recently refined these 'artificial selection' approaches to focus on microorganisms. The hope is to obtain microbes equipped with desirable features, such as the ability to degrade plastic or to produce valuable molecules. However, existing ways of using artificial selection on microbes are limited and sometimes not effective.

Computer scientists have also harnessed evolutionary principles for their own purposes, developing highly effective artificial selection protocols that are used to find solutions to challenging computational problems. Yet because of limited communication between the two fields, sophisticated selection protocols honed over decades in evolutionary computing have yet to be evaluated for use in biological populations.

In their work, Lalejini et al. compared popular artificial selection protocols developed for either evolutionary computing or work with microorganisms. Two computing selection methods showed promise for improving directed evolution in the laboratory. Crucially, these selection protocols differed from conventionally used methods by selecting for both diversity and performance, rather than performance alone. These promising approaches are now being tested in the laboratory, with potentially far-reaching benefits for medical, biotech, and agricultural applications.

While evolutionary computing owes its origins to our understanding of biological processes, it has much to offer in return to help us harness those same mechanisms. The results by Lalejini et al. help to bridge the gap between computational and biological communities who could both benefit from increased collaboration.

techniques, and culturing devices. These advances have cultivated growing interest in directing the evolution of whole microbial communities with functions that can be harnessed in medical, biotech, and agricultural domains (*Sánchez et al., 2021*).

Attempting to direct evolutionary processes for applied goals has not been limited to biological systems. The field of *evolutionary computing* harnesses the principles of Darwinian evolution as a general-purpose search engine to find solutions to challenging computational and engineering problems (*Fogel, 2000*). As in evolutionary computing, directed evolution in the laboratory begins with a library—or population—of variants (e.g., communities, genomes, or molecules). Variants are scored based on a phenotypic trait (or set of traits) of interest, and the variants with the 'best' traits are chosen to produce the next generation. Such approaches to picking progenitors are known as *elitist* selection algorithms in evolutionary computing (*Baeck et al., 1997*). Notably, evolutionary computing research has shown that these elitist approaches to artificial selection can be suboptimal in complex search spaces. On their own, elitist selection schemes fail to maintain diversity, which can lead to populations becoming trapped on suboptimal regions of the search space because of a loss of variation for selection to act on (*Lehman and Stanley, 2011a*; *Hernandez et al., 2022b*). Elitist selection schemes also inherently lack mechanisms to balance selection across multiple objectives. These observations suggest that other approaches to selection may improve directed microbial evolution outcomes. Fortunately, artificial selection methods (i.e., parent selection algorithms or selection schemes) are intensely studied in evolutionary computing, and many in silico selection techniques have been developed that improve the quality and diversity of evolved solutions (e.g., *Spector, 2012*; *Mouret and Clune, 2015*; *Hornby, 2006*; *Goldberg and Richardson, 1987*; *Goings et al., 2012*; *Lehman and Stanley, 2011b*).

Given their success, we expect that artificial selection methods developed for evolutionary computing will improve the efficacy of directed microbial evolution in the laboratory, especially when simultaneously selecting for more than one trait (a common goal in evolutionary computation). Such techniques may also be useful in the laboratory to simultaneously select for multiple functions of interest, different physical and growth characteristics, robustness to perturbations, or the ability to grow in a range of environments. Directed microbial evolution, however, differs from evolutionary computing in ways that may inhibit our ability to predict which techniques are most appropriate

for the laboratory. For example, candidate solutions (i.e., individuals) in evolutionary computing are evaluated one by one, resulting in high-resolution genotypic and phenotypic information that can be used for selecting parents, which are then copied, recombined, and mutated to produce offspring. In directed microbial evolution, individual-level evaluation is generally intractable at scales required for de novo evolution; as such, evaluation often occurs at the population level, and the highest performing populations are partitioned (instead of copied) to create 'offspring' populations. Moreover, when traits of interest do not benefit individuals' reproductive success, population-level artificial selection may conflict with individual-level selection, which increases the difficulty of steering evolution.

Here, we ask whether artificial selection techniques developed for evolutionary computing might be useful for directing the evolution of microbial populations when selecting for multiple traits of interest. We examine selection both for enhancing multiple traits in a single microbial strain and for producing a set of diverse strains that each specialize in different traits. To do so, we developed an agent-based model of directed evolution wherein we evolve populations of self-replicating computer programs performing computational tasks that contribute either to the phenotype of the individual or the phenotype of the population. Using our model, we evaluated how well three selection techniques from evolutionary computing (tournament, lexicase, and non-dominated elite selection) performed in a setting that mimics directed evolution on functions measurable at the population level. Tournament selection chooses progenitors by selecting the most performant candidates in each of a series of randomly formed 'tournaments.' Both lexicase and non-dominated elite selection focus on propagating a diverse set of candidates that balance multiple objectives in different ways. These selection techniques are described in detail in the 'Methods' section.

Overall, we found that multiobjective selection techniques (lexicase and non-dominated elite selection) generally outperformed the selection schemes commonly applied to directed microbial evolution (elite and top 10%). In particular, our findings suggest that lexicase selection is a good candidate technique to translate into the laboratory, especially when aiming to evolve a diverse set of specialist microbial populations. Additionally, we found that population-level artificial selection can improve directed evolution outcomes even when traits of interest are directly selected (i.e., the traits are correlated with individual-level reproductive success).

These findings lay the foundation for strengthened communication between the evolutionary computing and directed evolution communities. The evolution of biological organisms (both natural and artificial) inspired the origination of evolutionary computation, and insights from evolutionary biology are regularly applied to evolutionary computing. As evolutionary computation has immense potential as a system for studying how to control laboratory evolution, these communities are positioned to form a virtuous cycle where insights from evolutionary computing are then applied back to directing the evolution of biological organisms. With this work, we seek to strengthen this feedback loop.

## Directed evolution

Humans have harnessed evolution for millennia, applying artificial selection (knowingly and unknowingly) to domesticate a variety of animals, plants, and microorganisms (*Hill and Caballero, 1992*; *Cobb et al., 2013*; *Driscoll et al., 2009*; *Libkind et al., 2011*). More recently, a deeper understanding of evolution, genetics, and molecular biology in combination with technological advances has extended the use of artificial selection beyond domestication and conventional selective breeding. For example, artificial selection has been applied to biomolecules (*Beaudry and Joyce, 1992*; *Chen and Arnold, 1993*; *Esvelt et al., 2011*), genetic circuits (*Yokobayashi et al., 2002*), microorganisms (*Ratcliff et al., 2012*), viruses (*Burrowes et al., 2019*; *Maheshri et al., 2006*), and whole microbial communities (*Goodnight, 1990*; *Swenson et al., 2000*; *Sánchez et al., 2021*). In this work, we focus on directed microbial evolution.

One approach to artificial selection is to configure organisms' environment such that desirable traits are linked to growth or survival (referred to as 'selection-based methods'; *Wang et al., 2021*). In some sense, these selection-based methods passively harness artificial selection as individuals with novel or enhanced functions of interest will tend to outcompete other conspecifics without requiring intervention beyond initial environmental manipulations. In combination with continuous culture devices, this approach to directing evolution can achieve high-throughput microbial directed evolution, 'automatically' evaluating many variants without manual analysis (*Wang et al., 2021*; *Toprak*

*et al., 2012*; *DeBenedictis et al., 2021*). For example, to study mechanisms of antibiotic resistance, researchers have employed morbidostats that continuously monitor the growth of evolving microbial populations and dynamically adjust antibiotic concentrations to maintain constant selection on further resistance (*Toprak et al., 2012*). However, linking desirable traits to organism survival can be challenging, requiring substantial knowledge about the organisms and the functions of interest.

Similar to conventional evolutionary algorithms, 'screening-based methods' of directed evolution assess each variant individually and choose the most promising to propagate (*Wang et al., 2021*). Overall, screening-based methods are more versatile than selection-based methods because traits that are desirable can be directly discerned. However, screening requires more manual intervention and thus limits throughput. In addition to their generality, screening-based methods also allow practitioners to more easily balance the relative importance of multiple objectives. For example, plant breeders might simultaneously balance screening for yield, seed size, drought tolerance, etc. (*Cooper et al., 2014*; *Bruce et al., 2019*).

In this work, we investigate screening-based methods of directed microbial evolution as many insights and techniques from evolutionary computation are directly applicable. When directing microbial evolution, screening is applied at the population (or community) level (*Xie and Shou, 2021*; *Sánchez et al., 2021*). During each cycle of directed microbial evolution, newly founded populations grow over a maturation period in which members of each population reproduce, mutate, and evolve. Next, populations are assessed, and promising populations are chosen as 'parental populations' that will be partitioned into the next generation of 'offspring populations.'

Screening-based artificial selection methods are analogous to parent selection algorithms or *selection schemes* in evolutionary computing. Evolutionary computing research has shown that the most effective selection scheme depends on a range of factors, including the number of objectives (e.g., single- versus multiobjective), the form and complexity of the search space (e.g., smooth versus rugged), and the practitioner's goal (e.g., generating a single solution versus a suite of different solutions). Conventionally, however, screening-based methods of directing microbial evolution choose the overall 'best'-performing populations to propagate (e.g., the single best population or the top 10%; *Xie et al., 2019*).

To the best of our knowledge, the more sophisticated methods of choosing progenitors from evolutionary computing have not been applied to directed evolution of microbes. However, artificial selection techniques from evolutionary computing have been applied in a range of other biological applications. For example, multiobjective evolutionary algorithms have been applied to DNA sequence design (*Shin et al., 2005*; *Chaves-González, 2015*); however, these applications are treated as computational optimization problems. A range of selection schemes from evolutionary computing have also been proposed for both biomolecule engineering (*Currin et al., 2015*; *Handl et al., 2007*) and agricultural selective breeding (especially for scenarios where genetic data can be exploited) (*Ramasubramanian and Beavis, 2021*). For example, using an NK landscape model, O'Hagan et al. evaluated the potential of elite selection, tournament selection, fitness sharing, and two rule-based learning selection schemes for selective breeding applications (*O'Hagan et al., 2012*). Inspired by genetic algorithms, island model approaches (*Tanese, 1989*) have been proposed for improving plant and animal breeding programs (*Ramasubramanian and Beavis, 2021*; *Yabe et al., 2016*), and *Akdemir et al., 2019* applied multiobjective selection algorithms like non-dominated selection to plant and animal breeding. In each of these applications, however, artificial selection acted as screens on individuals and not whole populations; therefore, our work focuses on screening at the population level in order to test the applicability of evolutionary computing selection algorithms as general-purpose screening methods for directed microbial evolution.

## Methods

Conducting directed evolution experiments in the laboratory can be slow and labor intensive, making it difficult to evaluate and tune new approaches to artificial selection in vitro. We could draw directly from evolutionary computing results when transferring techniques into the laboratory, but the extent to which these results would predict the efficacy (or appropriate parameterization) of a given algorithm in a laboratory setting is unclear. To fill this gap, we developed an agent-based model of directed evolution of microbes for evaluating which techniques from evolutionary computing might be most applicable in the laboratory.

**Table 1.** Computational functions that conferred individual-level or population-level benefits. The particular functions were chosen to be used in our model based on those used in the Avida system (**Bryson et al., 2021**). In all experiments, we included two versions of ECHO (each for different input values), resulting in 22 possible functions that organisms could perform. In general, functions that confer population-level benefits are more complex (i.e., require more instructions to perform) than functions designated to confer individual-level benefits.

| Function | # Inputs | Benefit |
|---|---|---|
| ECHO | 1 | Individual |
| NAND | 2 | Individual |
| NOT | 1 | Population |
| ORNOT | 2 | Population |
| AND | 2 | Population |
| OR | 2 | Population |
| ANDNOT | 2 | Population |
| NOR | 2 | Population |
| XOR | 2 | Population |
| EQU | 2 | Population |
| $2A$ | 1 | Individual |
| $A^2$ | 1 | Population |
| $A^3$ | 1 | Population |
| $A + B$ | 2 | Population |
| $A \times B$ | 2 | Population |
| $A - B$ | 2 | Population |
| $A^2 + B^2$ | 2 | Population |
| $A^3 + B^3$ | 2 | Population |
| $A^2 - B^2$ | 2 | Population |
| $A^3 - B^3$ | 2 | Population |
| $\frac{A+B}{2}$ | 2 | Population |

Using our model of laboratory directed evolution, we investigated whether selection schemes from evolutionary computing might be useful for directed evolution of microbes. Specifically, we compared two selection schemes used in directed evolution (elite and top 10% selection) with three other methods used in evolutionary computing (tournament, lexicase, and non-dominated elite selection). Additionally, we ran two controls that ignored population-level performance.

We conducted three independent experiments. First, we evaluated the relative performance of parent selection algorithms in a conventional evolutionary computing context, which established baseline expectations for subsequent experiments using our model of laboratory directed evolution. Next, we compared parent selection algorithms using our model of laboratory directed evolution in two contexts. In the first context, we did not link population-level functions (**Table 1**) to organism survival to evaluate how well each parent selection algorithm performs as a screening-based method of artificial selection. In the second context, we tested whether any of the selection schemes still improve overall directed evolution outcomes even when organism survival *is* aligned with population-level functions.

## Digital directed evolution

**Figure 1** overviews our model of laboratory directed microbial evolution. Our model contains a set of populations (i.e., a 'metapopulation'). Each population comprises digital organisms (self-replicating computer programs) that compete for space in a well-mixed virtual environment. Both the digital organisms and their virtual environment are inspired by those of the Avida Digital Evolution Platform (**Ofria et al., 2009**), which is a well-established study system for in silico evolution experiments (e.g., **Lenski et al., 1999**; **Lenski et al., 2003**; **Zaman et al., 2014**; **Lalejini et al., 2021**) and is a closer analog to microbial evolution than conventional evolutionary computing systems. However, we note that our model's implementation is fully independent of Avida, as the Avida software platform does not allow us to model laboratory setups of directed microbial evolution (as described in the previous section).

In our model, we seed each population with a digital organism (explained in more detail below) capable only of self-replication (**Figure 1a**). After initialization, directed evolution proceeds in cycles. During a cycle, we allow all populations to evolve for a fixed number of time steps (i.e., a 'maturation period'; **Figure 1b**). During a population's maturation period, digital organisms execute the computer code in their genomes, which encodes the organism's ability to self-replicate and perform computational tasks using inputs from its environment. When an organism reproduces, its offspring is subject to mutation, which may affect its phenotype. Therefore, each population in the metapopulation

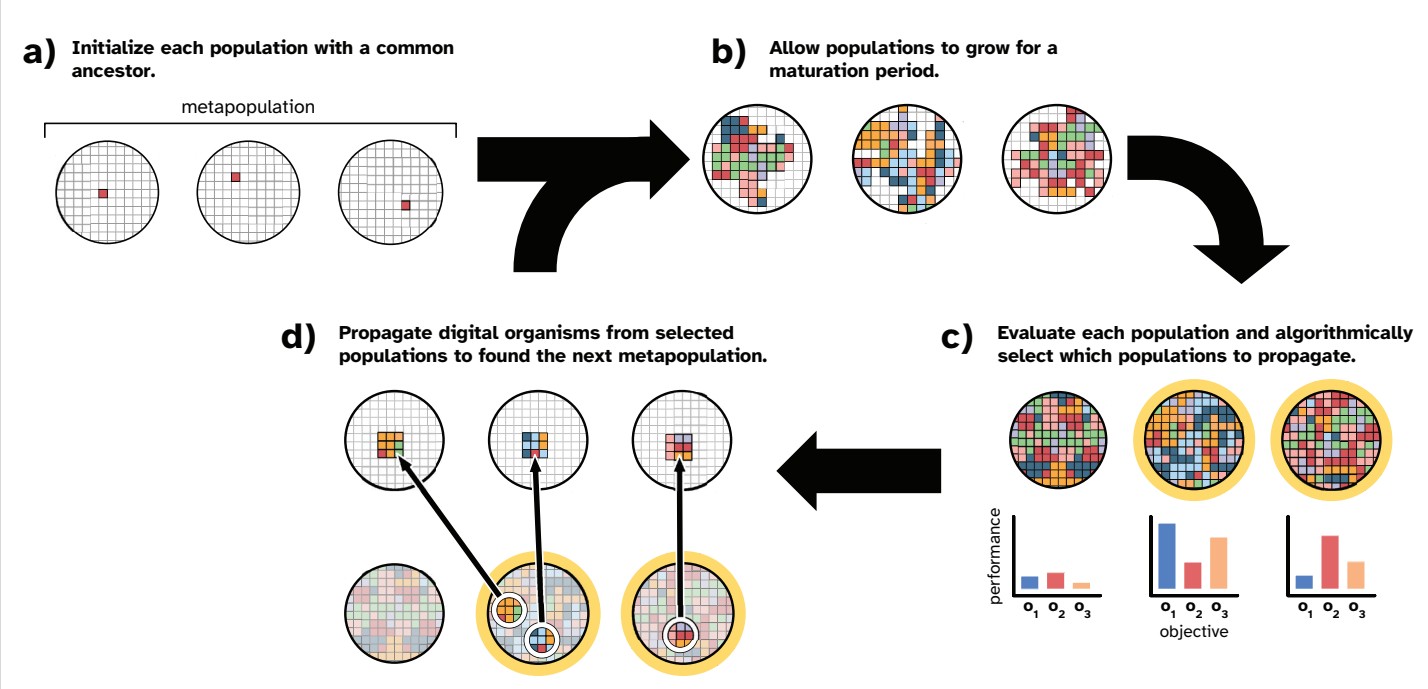

**Figure 1.** Overview of our model of directed microbial evolution. In (**a**), we found each of $N$ populations with a single digital organism. In this figure, the metapopulation comprises three populations. Next (**b**), each population undergoes a maturation period during which digital organisms compete for space, reproduce, mutate, and evolve. After maturation, (**c**) we evaluate each population based on one or more population-level characteristics, and we select populations (repeat selections allowed) to partition into $N$ 'offspring' populations. In this figure, we show populations being evaluated on three objectives ($o_1$, $o_2$, and $o_3$). In this work, population-level objectives include the ability to compute different mathematical expressions (see **Table 1**). We see this as analogous to a microbial population's ability to produce different biomolecules or to metabolize different resources. After evaluation, populations are chosen algorithmically using one of the selection protocols described in 'Methods.'

independently evolves during the maturation period. After the maturation period, we evaluate each population's performance on a set of objectives and apply an artificial selection protocol to algorithmically choose performant populations to propagate (**Figure 1c**).

In this work, we simulate a serial batch culture protocol. To create an offspring population (**Figure 1d**), we use a random sample of digital organisms from the chosen parental population (here we used 1% of the maximum population size). We chose this sample size based on preliminary experiments, wherein we found that smaller sample sizes performed better than larger sizes (see supplemental material, **Lalejini et al., 2022**).

## Digital organisms

Each digital organism contains a sequence of program instructions (its genome) and a set of virtual hardware components used to interpret and express those instructions. The virtual hardware and genetic representation used in this work extends that of **Dolson et al., 2019**; **Hernandez et al., 2022a**. The virtual hardware includes the following components: an instruction pointer indicating the position in the genome currently being executed, 16 registers for performing computations, 16 memory stacks, input and output buffers, 'scopes' that facilitate modular code execution, and machinery to facilitate self-copying. For brevity, we refer readers to supplemental material for a more detailed description of these virtual hardware components (**Lalejini et al., 2022**).

Digital organisms express their genomes sequentially unless the execution of one instruction changes which instruction should be executed next (e.g., 'if' instructions). The instruction set is Turing complete and syntactically robust such that any ordering of instructions is valid (though not necessarily useful). The instruction set includes operators for basic math, flow control (e.g., conditional logic and looping), designating and triggering code modules, input, output, and self-replication. Each

instruction contains three arguments, which may modify the effect of the instruction, often specifying memory locations or fixed values. We further document the instruction set in our supplemental material.

Digital organisms reproduce asexually by executing copy instructions to replicate their genome one instruction at a time and then finally issuing a divide command. However, copying is subject to errors, including single-instruction and single-argument substitution mutations. Each time an organism executes a copy, there is a 1% chance that a random instruction is copied instead, introducing a mutation (and a 0.5% chance to incorrectly copy each instruction argument). Mutations can change the offspring's phenotype, including its replication efficiency and computational task repertoire.

Genomes were fixed at a length of 100 instructions. When an organism replicates, its offspring is placed in a random position within the same population, replacing any previous occupant. We limited the maximum population size to 1000 organisms. Because space is a limiting resource, organisms that replicate quickly have a selective advantage within populations.

During evolution, organism replication can be improved in two ways: by improving computational efficiency or by increasing the rate of genome execution ('metabolic rate'). An organism's metabolic rate determines the average number of instructions an organism is able to execute in a single time step. Digital organisms can improve their metabolic rate by evolving the ability to perform designated functions (referred to as individual-level functions), including some Boolean logic functions and simple mathematical expressions (*Table 1*).

Performing a function requires the coordinated execution of multiple genetically encoded instructions, including ones that interact with the environment, store intermediate computations, and output the results. For example, the $A + B$ function requires an organism to execute the input instruction twice to load two numeric inputs into its memory registers, execute an add instruction to sum those two inputs and store the result, and then execute an output instruction to output the result.

When an organism produces output, we check to see whether the output completes any of the designated functions (given previous inputs it received); if so, the organism's metabolic rate is adjusted accordingly. Organisms are assigned a random set of numeric inputs at birth that determine the set of values accessible via the input instruction. We guarantee that the set of inputs received by an organism result in a unique output for each designated function. Organisms benefit from performing each function only once, preventing multiple rewards for repeating a single-function result. In this work, we configured each function that confers an individual-level benefit to double an organism's metabolic rate, which doubles the rate the organism can copy itself. For a more in-depth overview of digital organisms in a biological context, see *Wilke and Adami, 2002*.

## Population-level evaluation

In addition to individual-level functions, organisms can perform 18 different population-level functions (*Table 1*). Unless stated otherwise, performing a population-level function does not improve an organism's metabolic rate. Instead, population-level functions are used for population-level evaluation and selection, just as we might screen for the production of different by-products in laboratory populations. We assigned each population a score for each population-level function based on the number of organisms that performed that function during the population's maturation period. The use of these scores for selecting progenitors varied by selection scheme (as described in the 'Methods section).

While population-level functions benefit a population's chance to propagate, they do not benefit an individual organism's immediate reproductive success: time spent computing population-level functions is time not spent on performing individual-level functions or self-replicating. Such conflicts between group-level and individual-level fitness are well-established in evolving systems (*Simon et al., 2013*; *Waibel et al., 2009*) and are indeed a problem recognized for screening-based methods of artificial selection that must be applied at the population level (*Escalante et al., 2015*; *Brenner et al., 2008*).

## Selection schemes

### Elite and top 10% selection

Elite and top 10% selection are special cases of truncation selection (*Mühlenbein and Schlierkamp-Voosen, 1993*) or $(\mu, \lambda)$ evolutionary strategies (*Bäck et al., 1991*) wherein candidates are ranked and the most performant are chosen as progenitors. We implement these selection methods as they are

often used in laboratory directed evolution (*Xie et al., 2019*; *Xie and Shou, 2021*). Here, both elite and top 10% selection rank populations according to their aggregate performance on all population-level functions. Elite selection chooses the single best-performing population to generate the next metapopulation, and top 10% chooses the best 10% (rounded up to the nearest whole number) as parental populations.

## Tournament selection

Tournament selection is one of the most common parent selection methods in evolutionary computing. To select a parental population, $T$ populations are randomly chosen (with replacement) from the metapopulation to form a tournament ($T = 4$ in this work). The population with the highest aggregate performance on all population-level functions wins the tournament and is chosen as a parent. We run $N$ tournaments to choose the parental populations for each of $N$ offspring populations.

## Lexicase selection

Unlike the previously described selection schemes, lexicase selection does not aggregate measures of performance across population-level functions (i.e., objectives) to choose parental populations. Instead, lexicase selection considers performance on each population-level function independently. For each parent selection event, all members of the metapopulation are initially considered candidates for selection. To select a parental population, the set of population-level functions is shuffled and each function is considered in sequence. Each function (in shuffled order) is used to filter candidates sequentially, removing all but the best candidates from further consideration. This process continues until only one candidate remains to be selected or until all functions have been considered; if more than one candidate remains, one is selected at random.

Lexicase selection was originally proposed for test-based genetic programming problems (*Spector, 2012*; *Helmuth and Spector, 2015a*), but has since produced promising results in a variety of domains (*Moore and Stanton, 2017*; *La Cava et al., 2016*; *Metevier et al., 2019*; *Aenugu and Spector, 2019*). By randomly permuting the objectives for each parent selection, lexicase selection maintains diversity (*Dolson et al., 2018*; *Helmuth et al., 2016*), which improves search space exploration (*Hernandez et al., 2022b*) and overall problem-solving success (*Hernandez et al., 2022a*; *Helmuth et al., 2015b*). In particular, lexicase selection focuses on maintaining specialists (*Helmuth et al., 2019*).

## Non-dominated elite selection

Non-dominated elite selection is a simple multiobjective selection algorithm that chooses all populations that are not Pareto dominated by another population (*Zitzler, 1999*). A candidate, $c_a$, Pareto dominates another candidate, $c_b$, if the following two conditions are met: (1) $c_a$ performs no worse than $c_b$ on all population-level functions and (2) $c_a$ has strictly better performance than $c_b$ on at least one population-level function. The set of all non-dominated populations is then sampled with replacement to seed offspring populations.

Pareto domination is a fundamental component in many successful evolutionary multiobjective optimization (EMOO) algorithms (*Fonseca and Fleming, 1995*; *Zitzler, 1999*; *Horn et al., 1994*; *Deb et al., 2002*). In general, EMOO algorithms aim to produce the set of solutions with optimal trade-offs of the objective set. Most EMOO algorithms have more sophisticated routines for parent selection than non-dominated elite selection (e.g., use of external archives or crowding metrics). We opted to use non-dominated elite selection for its simplicity, but future work will explore more EMOO selection schemes.

## Selection controls

We used random and no selection treatments as controls. Random selection chooses a random set of populations (with replacement) to serve as parental populations. 'No selection' chooses all populations in the metapopulation as sources for founding the next generation of populations; that is, each population is chosen to produce one offspring population. Neither of these controls apply selection pressure for performing population-level functions.

## Experimental design

### Establishing baseline problem-solving expectations in an evolutionary computing context

First, we evaluated the relative performance of parent selection algorithms in a conventional evolutionary computing context (linear genetic programming; *Brameier and Banzhaf, 2007*), in which we evolved programs to compute the functions in *Table 1*. This control experiment followed conventional evolutionary computing setup: there is no metapopulation, and programs are evaluated, selected, and propagated as individuals. We used this control experiment to verify that the genetic representation used by digital organisms is sufficient for evolving each of the computational functions used in subsequent experiments. Additionally, the relative performances of each algorithm establishes an expectation for how each parent selection algorithm might perform in our model of laboratory directed evolution.

For each of the seven selection schemes described previously, we evolved 50 replicate populations of 1000 programs. We initialized each replicate population with 1000 randomly generated programs. In evolution experiments with microbes (in silico or in vitro), the number of elapsed organism generations is measured (or estimated). However, in conventional evolutionary computing experiments, generations are configured by the experimenter and determine the duration of the experiment. We chose to evolve these populations for 55,000 generations, which approximates the elapsed number of organism-level generations measured after 2000 cycles of population-level artificial selection in exploratory runs of simulated directed evolution.

Each generation, we independently evaluated each individual to determine its phenotype. To evaluate a program, we executed it for 200 time steps (i.e., the program could execute 200 instructions) and tracked the program's inputs and outputs to determine which of the functions in *Table 1* it performed (if any). We used the same genetic representation used by digital organisms in our model of simulated directed evolution. However, we excluded self-replication instructions from the instruction set as we did not require programs to copy themselves during this experiment.

After evaluating each individual, we selected 'parents' to contribute offspring to the next generation. For the purpose of selection, we treated each of the 22 possible functions as a pass–fail task. Lexicase and non-dominated elite selection considered each task separately to choose parent programs, while elite, top 10%, and tournament selection used the number of task passes as fitness values for choosing parents. Chosen parents reproduced asexually, and we applied mutations to offspring of the same types and frequencies as in our model of laboratory directed evolution.

At the end of each run, we identified the individual program that performed the most tasks, and we used these programs to compare performance across treatments. Adopting conventional evolutionary computation metrics, we considered a replicate population to be *successful* if it produced a program capable of performing all 22 tasks during evaluation.

### Applying parent selection algorithms in a digital directed evolution context

Next, we evaluated each selection scheme's performance in our model of laboratory directed evolution. For each selection scheme, we ran 50 independent replicates of directed evolution for 2000 cycles of population maturation, screening, and propagation (as shown in *Figure 1*). During each cycle, we gave populations a maturation period of 200 updates (one update is the amount of time required for the average organism in a population to execute 30 instructions). Each 200-update maturation period allowed for approximately 25–35 generations of evolution, resulting in a total of approximately 50,000–70,000 organism-level generations after 2000 cycles of simulated laboratory directed evolution. Within each replicate, the metapopulation was composed of 96 populations (following the number of samples held by a standard microtiter plate used in laboratory experiments), each with a maximum carrying capacity of 1000 digital organisms. During a population's maturation period, we measured the number of organisms that performed each of the 18 population-level functions (*Table 1*) as the population's 'phenotype' for evaluation. We selected populations to propagate according to the treatment-specific selection scheme and propagated chosen parental populations.

At the end of the experiment, we analyzed the population-level functions performed by populations in each replicate's metapopulation. First, we calculated each population's 'task profile,' which is a binary vector that describes which population-level functions are 'covered' by the population (zeroes are assigned for functions that are not covered and ones for those that are covered). A function is

considered covered if it is performed by at least 50 organisms (a threshold ensuring the performance was not one-off) during a given maturation period.

Next, we measured the 'best population task coverage' and 'metapopulation task coverage' for each replicate. Best population task coverage is measured as the number of functions covered by the population with the largest set of covered functions. Metapopulation task coverage is measured as the number of functions covered across the entire metapopulation (i.e., the union of unique tasks covered by each population in the metapopulation).

We also measured the phenotypic diversity within each metapopulation. Specifically, we measured the number of different task profiles present in the metapopulation (i.e., phenotypic richness) and the 'spread' of task profiles in the metapopulation. To measure a metapopulation's task profile spread, we calculated a centroid task profile as the average of all task profiles in the metapopulation, and then we calculated the average normalized cosine distance between each population's task profile and the centroid. A metapopulation's task spread summarizes how different the constituent populations' task profiles are from one another.

## Evaluating whether selection schemes improve directed evolution outcomes when population-level functions are aligned with organism survival

Selection-based methods of artificial selection tie desired traits to organism survival, eliminating the need to apply screening-based methods to populations. We tested whether the addition of population-level selection improves directed evolution outcomes even when traits of interest (population-level functions) are selected for at the individual level (i.e., tied to organism survival). To do so, we repeated our previously described directed evolution experiment, except we configured all population-level functions to improve an organism's metabolic rate in addition to the individual-level functions. As such, all population-level functions were beneficial in all treatments, including the random and no selection controls. However, only treatments with non-control selection schemes applied artificial selection at the population level.

### Statistical analyses

In general, we differentiated between sample distributions using non-parametric statistical tests. For each major analysis, we first performed a Kruskal–Wallis test (*Kruskal and Wallis, 1952*) to determine whether there were significant differences in results across treatments (significance level $\alpha = 0.05$). If so, we applied a Wilcoxon rank-sum test (*Wilcoxon, 1992*) to distinguish between pairs of treatments using a Bonferroni correction for multiple comparisons (*Rice, 1989*). Due to space limitations, we do not report all pairwise comparisons in our main results; however, all of our statistical results are included in our supplemental material.

### Software and data availability

Our model of laboratory directed evolution is available on GitHub (see *Lalejini et al., 2022*) and is implemented using the Empirical scientific software library (*Ofria et al., 2020*). We conducted all statistical analyses with R version 4.04 (*R Development Core Team, 2021*), using the following R packages for data analysis and visualization: tidyverse (*Wickham et al., 2019*), ggplot2 (*Wickham, 2016*), cowplot (*Wilke, 2020*), viridis (*Garnier, 2018*), khroma (*Frerebeau, 2022*), and Color Brewer (*Harrower and Brewer, 2003*; *Neuwirth, 2014*). Our source code for experiments, analyses, and visualizations is publicly available on GitHub (see *Lalejini et al., 2022*). Additionally, our experiment data are publicly archived on the Open Science Framework (see *Lalejini, 2022*).

## Results and discussion
### Baseline problem-solving expectations in an evolutionary computing context

First, we established baseline performance expectations for the selection schemes in a conventional genetic programming context to validate the solvability of the individual- and population-level functions used in our digital directed evolution experiments. Two selection schemes produced successful replicates, where success is defined as evolving a program capable of performing all 22 functions: elite (1/50) and lexicase selection (47/50). No replicates solved this set of tasks in any other treatment.

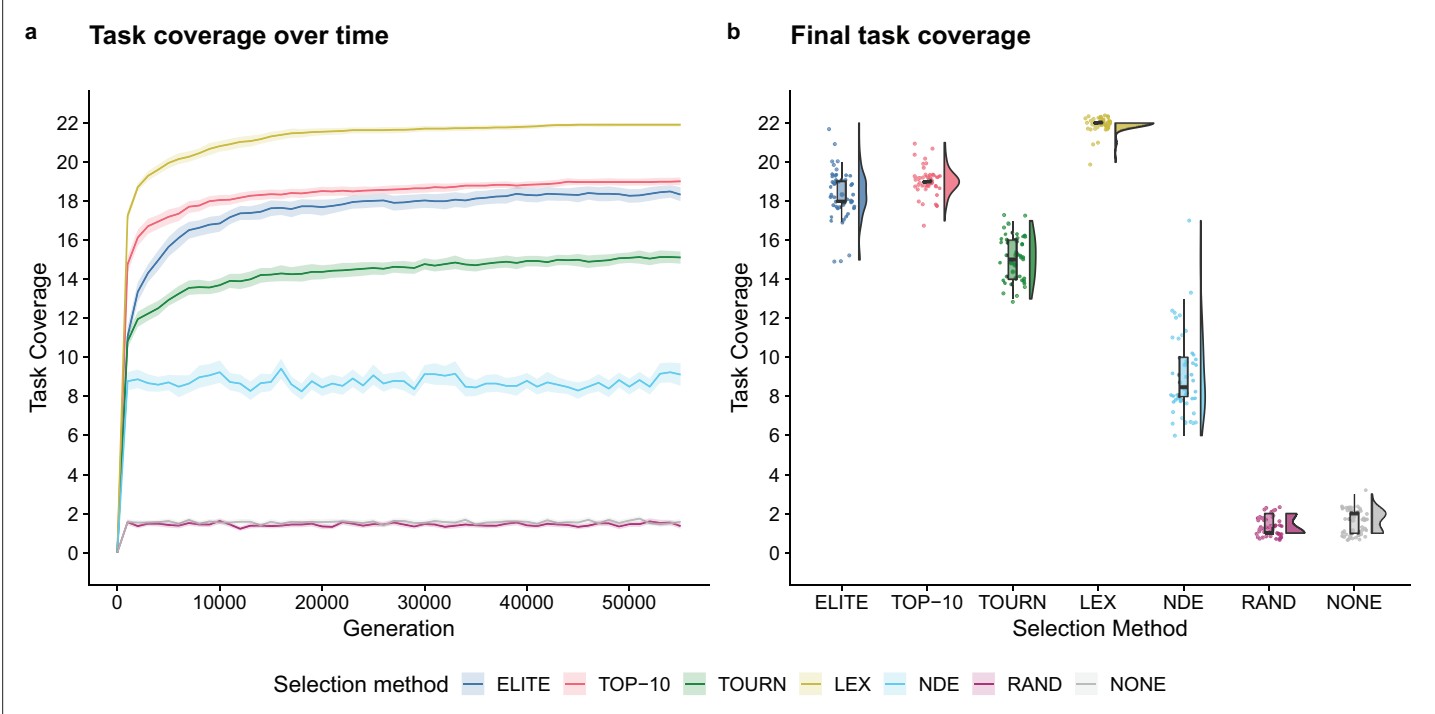

**Figure 2.** Task coverage of the best program (per replicate) evolved in an evolutionary computing context over time (**a**) and at the end of the experiment (**b**). Selection scheme abbreviations are as follows: TOURN, tournament; LEX, lexicase; NDE, non-dominated elite; RAND, random; NONE, no selection. In panel (**a**), each line gives the mean value across 50 replicates, and the shading around each mean gives a bootstrapped 95% confidence interval. Differences in final task coverage among treatments were statistically significant (Kruskal–Wallis, $p < 10^{-4}$).

*Figure 2* depicts the number of functions performed by the best program from each replicate (over time and at the end of the experiment). All selection schemes outperformed the random and no selection controls. At the end of the experiment, differences between all pairs except random and no selection were statistically significant (Bonferroni-corrected Wilcoxon rank-sum, p<0.01). Lexicase selection was the most performant followed by top 10%, elite, tournament, and non-dominated elite selection.

These data confirm that our genetic representation allows for the evolution of each computational function used in our model of laboratory directed evolution. Moreover, these data establish some expectations for the relative performance of each selection scheme in our directed evolution experiments. Lexicase selection's strong performance is consistent with previous work demonstrating its efficacy on program synthesis problems (*Helmuth and Spector, 2015a*; *Helmuth and Abdelhady, 2020*). While initially surprised by non-dominated elite's poor performance (relative to other non-control selection schemes), we note that selection methods based on Pareto domination are rarely applied to pass–fail test-based genetic programming problems, and perhaps the course-grained function scores (0 or 1) hindered its capacity for problem-solving success.

### Lexicase and non-dominated elite selection show promise for directed evolution

Next, we compared selection scheme performance when modeling the directed evolution of digital organisms. *Figure 3a–d* shows the best population and metapopulation task coverages. By the end of the experiment, all selection schemes resulted in greater single-population task coverage than both random and no selection controls (Bonferroni-corrected Wilcoxon rank-sum test, $p < 10^{-4}$). Metapopulation coverage under tournament selection was not significantly different than coverage under the no selection control, but all other selection schemes resulted in significantly better metapopulation coverage than both controls (Bonferroni-corrected Wilcoxon rank-sum, p<0.03). Overall, lexicase and non-dominated elite selection scored the greatest population and metapopulation task coverage out of all selection schemes, and lexicase was the overall best selection scheme according to both metrics of performance. Moreover, we found that lexicase selection improved both single-population and

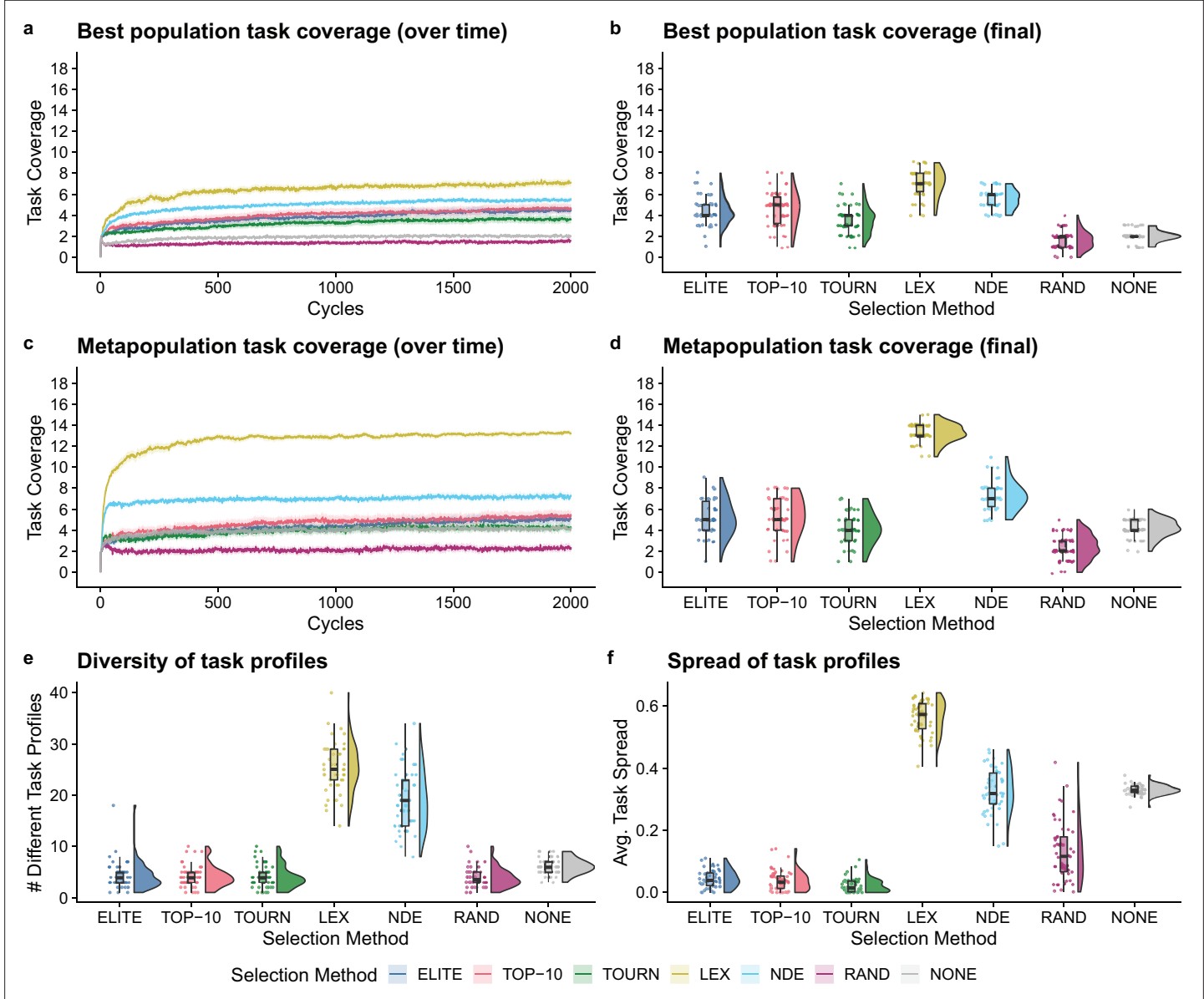

**Figure 3.** Digital directed evolution results. For panels (**a**) and (**c**), each line gives the mean value across 50 replicates, and the shading around each mean gives a bootstrapped 95% confidence interval. At the end of the experiment, differences among treatments were statistically significant for each of (**b**) best population task coverage, (**d**) metapopulation task coverage, (**e**) task profile diversity, and (**f**) task profile spread (Kruskal–Wallis, $p < 10^{-4}$).

metapopulation task coverage relative to all other selection methods after just 10 cycles of directed evolution (Bonferroni-corrected Wilcoxon rank-sum, $p < 0.04$).

While differences were significant in the best population task coverage, they were not necessarily substantial. However, other measures had more substantial differences. Both multiobjective selection schemes—lexicase and non-dominated elite—had the greatest metapopulation task coverage (*Figure 3d*) and the greatest diversity of task profiles in the final metapopulations (*Figure 3e*; Bonferroni-corrected Wilcoxon rank-sum test, $p < 10^{-4}$). Lexicase selection in particular also had the greatest task profile *spread* (*Figure 3f*; Bonferroni-corrected Wilcoxon rank-sum test, $p < 10^{-4}$), which is consistent with previous results demonstrating that lexicase excels at maintaining diverse specialists (*Dolson et al., 2018*; *Helmuth et al., 2019*; *Hernandez et al., 2022a*; *Helmuth et al., 2019*).

The lack of task profile diversity maintained by each of the elite, top 10%, and tournament selection methods is consistent with empirical and theoretical evolutionary computing studies (*Dolson et al., 2019*; *Hernandez et al., 2022c*). Similar results have also been observed in directed evolution

of bacterial communities. For example, after just six generations of selecting the top 25% of bacterial communities, Chang et al. observed that their entire metapopulation of bacterial communities descended from the initial top community, resulting in no functional variance for further selection to act on *Chang et al., 2020*. Indeed, the inability to reliably maintain diversity is an intrinsic limitation of elite, top 10%, and tournament selection, regardless of the substrate in which they are applied.

We hypothesized that lexicase and non-dominated elite selection's mechanisms for selecting different *types* of parental populations underpinned their improved performance over elite, top 10%, and tournament selection. This, however, is confounded by each selection scheme's varying capacity to select a greater number of different populations (regardless of differences in those selected). As such, we asked whether lexicase and non-dominated elite's success could be explained by a capacity to select a greater number of different parental populations. Elite selection selected exactly one population per cycle, top 10% selected 10, lexicase selected an average of 12, tournament selected an average of 50, and non-dominated elite selected an average of 83 different populations. Thus, we can rule out the number of populations selected per cycle as the sole explanation for lexicase selection's success; we argue that this, in combination with our diversity data, suggests that directed evolution practitioners should consider incorporating mechanisms for selecting phenotypically diverse parental populations into their artificial selection approaches.

Of all non-control selection methods, non-dominated elite exhibited the least selection pressure, selecting an average of 83 different populations (out of 96 possible) per cycle. This is a known problem for naive Pareto-based multiobjective selection methods (*He and Yen, 2016*; *Ibrahim et al., 2016*). For example, for a problem with more than five objectives, *He et al., 2014* found that over 90% of candidates in a randomly generated population were non-dominated. Our experiments corroborate this finding as non-dominated elite selection chose an average of nearly 90% of the metapopulation to be propagated each cycle. Evolutionary computing practitioners have developed many approaches to improve selection pressure under Pareto-based multiobjective selection methods that could be applied to laboratory microbial populations. For example, rather than selecting *all* non-dominated candidates (as we did in our experiments), many Pareto-based multiobjective selection methods focus selection pressure on a diverse set of non-dominated candidates that represent different regions of the Pareto front (*Coello Coello, 1999*; *Deb et al., 2000*; *He and Yen, 2016*). Yet, despite non-dominate elite's shortcomings, it performed well relative to conventional artificial selection methods used in directed microbial evolution, motivating future studies investigating the use of more sophisticated Pareto-based selection methods for directed microbial evolution.

These results are also informative when compared to our genetic programming control experiment (*Figure 2*). While results across these two contexts are not directly comparable, we found steering evolution at the population level to be more challenging than steering at the individual level (as in conventional evolutionary computing). For example, across all treatments, no single population in our model of directed evolution performed all 18 population-level functions. Yet, after a similar number of organism-level generations (~55,000), both elite and lexicase selection produced programs capable of all 22 functions in a genetic programming context. Even if we limited our genetic programming control experiment to only 2000 generations (matching the 2000 cycles of population-level artificial selection in our simulated directed evolution experiments), we found that conventional genetic programming produced more performant programs than those evolved under our model of laboratory directed evolution (supplemental material; *Lalejini et al., 2022*). We also observed differences in the rank order of selection schemes between the genetic programming and simulated laboratory directed evolution experiments. For example, non-dominated elite selection performed poorly in a genetic programming context relative to the other non-control selection schemes; however, non-dominated elite outperformed all selection schemes except lexicase selection in our model of laboratory directed evolution. On its own, non-dominated elite's difference in performance is not surprising as non-dominated elite selection is not conventionally used for evolving computer programs where evaluation criteria are evaluated on a pass–fail basis. More broadly, however, we argue that these results highlight modeling as an important intermediate step when evaluating which techniques from evolutionary computing are likely to be effective in a laboratory setting.

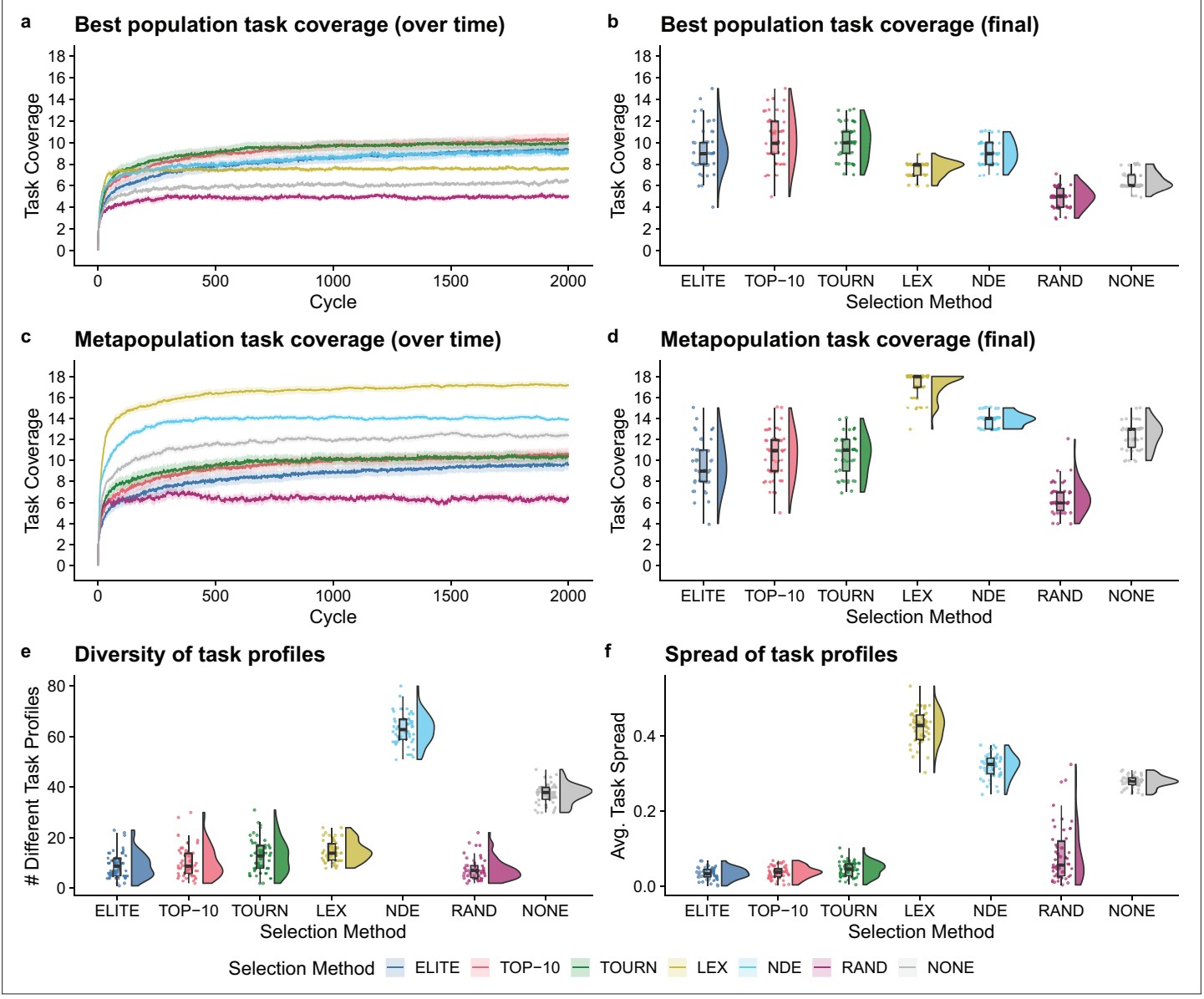

**Figure 4.** Digital directed evolution results when organism survival is tied to population-level functions. For panels (**a**) and (**c**), each line gives the mean value across 50 replicates, and the shading around each mean gives a bootstrapped 95% confidence interval. At the end of the experiment, differences among treatments were statistically significant for each of (**b**) best population task coverage, (**d**) metapopulation task coverage, (**e**) task profile diversity, and (**f**) task profile spread (Kruskal-Wallis, $p < 10^{-4}$).

## Selection schemes improve outcomes even when organism survival can be tied to population-level functions

Next, we tested whether the addition of population-level screening improves directed evolution outcomes even when population-level functions can be tied to organism survival. After 2000 cycles of directed evolution, each non-control selection scheme resulted in better single-population task coverage than either control treatment (*Figure 4b*; Bonferroni-corrected Wilcoxon rank-sum test, $p < 10^{-4}$). We did not find significant differences in best population coverage among elite, top 10%, tournament, and non-dominated elite selection. In contrast to our previous experiment, lexicase selection resulted in lower best population coverage than each other non-control selection scheme (Bonferroni-corrected Wilcoxon rank-sum test, $p < 10^{-4}$).

Lexicase selection, however, outperformed all other selection schemes on metapopulation task coverage (*Figure 4d*; Bonferroni-corrected Wilcoxon rank-sum test, $p < 10^{-4}$), producing 30

metapopulations that cover all 18 population-level functions. Even after just 10 cycles of directed evolution, lexicase selection improved metapopulation task coverage more than all other selection methods except non-dominated elite (Bonferroni-corrected Wilcoxon rank-sum test, $p<10^{-4}$). In general, lexicase selection produced metapopulations containing distinct specialist populations, resulting in high metapopulation task coverage while each specialist population had low task coverage on its own. Indeed, while lexicase metapopulations did not necessarily comprise many different population task profiles (*Figure 4e*), the task profiles were very different from one another (*Figure 4f*).

Of our two control selection methods, we found that performing no selection was better than random selection for both single-population and metapopulation task coverage (Bonferroni-corrected Wilcoxon rank-sum test, $p<10^{-4}$). In fact, performing no selection at all resulted in better metapopulation task coverage than elite, top 10%, and tournament selection (Bonferroni-corrected Wilcoxon rank-sum test, $p<10^{-3}$). We hypothesize that this result is because elite, top 10%, and tournament selection drive metapopulation convergence to homogeneous task profiles, while performing no selection at all allows populations to diverge from one another.

## Conclusion

In this work, we investigated whether the selection schemes from evolutionary computing might be useful for directing the evolution of microbial populations. To do so, we introduced an agent-based model of laboratory directed evolution. Overall, our results suggest that lexicase and non-dominated elite selection are promising techniques to transfer into the laboratory when selecting for multiple traits of interest as both of these selection schemes resulted in improved outcomes relative to conventional directed evolution selection methods. In some cases, these more sophisticated selection protocols improved directed evolution outcomes (relative to conventional methods) after just 10 rounds of artificial selection, which is especially promising given the effort required to carry out a single round of selection in the laboratory. We expect lexicase selection to be particularly useful for evolving a set of microbial populations, each specializing in different population-level functions. We also found that the addition of screening-based selection methods developed for evolutionary computation can improve directed evolution outcomes even in cases where organisms' reproductive success can be tied directly to the traits of interest.

Our study has several important limitations that warrant future model development and experimentation. For example, we focused on modeling microbial populations that grow (and evolve) in a simple environment without complex ecological interactions. We plan to add ecological dynamics by incorporating features such as limited resources, waste by-products, symbiotic interactions, and spatial structure. These extensions will allow us to model the directed evolution of complex microbial communities (e.g., *Chang et al., 2021*; *Xie and Shou, 2021*), which is an emerging frontier in laboratory directed evolution.

Additionally, our study models a serial batch culture laboratory setup wherein each offspring population is founded from a small sample of its parent population. Future work should investigate how different laboratory setups influence directed evolution outcomes across artificial selection protocols. For example, population bottlenecks are known to influence adaptation (*Izutsu et al., 2021*; *Mahrt et al., 2021*). Indeed, in exploratory experiments, we found that our choice of sample size influenced directed evolution outcomes under some selection protocols (see supplemental material; *Lalejini et al., 2022*). Other basic decisions including metapopulation size and the length of maturation period may also produce different outcomes under different artificial selection protocols. Future studies of how different artificial selection protocols perform under different laboratory conditions will help practitioners choose the most appropriate set of protocols given their particular constraints.

In this study, we compared simple versions of each selection scheme, and we did not tune their parameterizations in the context of our model of directed evolution. For example, we configured truncation selection (elite and top 10% selection) to reflect artificial selection methods commonly used in the laboratory, but adjusting the percentage of top populations chosen to propagate could improve directed evolution outcomes. Likewise, we expect that tournament selection's performance could be slightly improved by tuning the tournament size. However, we do not expect such tuning to change our overall findings as no amount of tuning would give these selection methods a mechanism to choose different *kinds* of populations as in lexicase and non-dominated elite selection.

We plan to test more sophisticated selection schemes as we continue to transfer techniques developed for evolutionary computation into the laboratory. For example, non-dominated elite selection is one of the simplest methods that uses Pareto domination to choose parents; given its strong performance, we see more sophisticated multiobjective selection algorithms (e.g., NSGA-II; *Deb et al., 2002*) as particularly promising for laboratory directed evolution. Lexicase selection variants are also promising for laboratory directed evolution: epsilon lexicase (*Spector et al., 2018*; *La Cava et al., 2016*) might be useful when population-level characteristics are measured as real-valued quantities, and cohort lexicase selection (*Hernandez et al., 2019*) could reduce the amount of screening required to select parental populations. Beyond selection schemes, we also see quality diversity algorithms (*Hagg, 2021*) as promising techniques to transfer into the laboratory (e.g., MAP-Elites; *Mouret and Clune, 2015*). Other techniques developed to improve evolutionary outcomes have analogues in both evolutionary computing and directed microbial evolution (often with independent intellectual origins), such as methods of perturbing populations to replenish variation (*Hornby, 2006*; *Chang et al., 2021*). These overlapping areas of research are fertile ground for mutually beneficial idea exchange between research communities.

We see digital experiments like the ones reported here as a critical step for transferring techniques developed for evolutionary computing into the laboratory. Indeed, our results are currently informing the design of laboratory experiments that apply evolutionary computing techniques to the directed evolution of *Escherichia coli*. Our model of directed microbial evolution provides a testbed for rigorously evaluating different artificial selection methods with different laboratory setups (e.g., metapopulation size, maturation period, etc.) before embarking on costly or time-consuming laboratory experiments.

## Acknowledgements

We thank the ZE[3] Laboratory for thoughtful discussions, feedback, and support. Additionally, we thank the reviewers for their thoughtful comments and suggestions. This research was supported through computational resources provided by the University of Michigan's Advanced Research Computing and by Michigan State University's Institute for Cyber-Enabled Research. Additionally, this research was supported by the National Science Foundation (NSF) through support to LZ (DEB-1813069) and a sub-award to AEV (MCB-1750125).

## Additional information

### Funding

| Funder | Grant reference number | Author |
| --- | --- | --- |
| National Science Foundation | DEB-1813069 | Luis Zaman |
| National Science Foundation | MCB-1750125 | Anya E Vostinar |

The funders had no role in study design, data collection and interpretation, or the decision to submit the work for publication.

### Author contributions

Alexander Lalejini, Conceptualization, Data curation, Software, Formal analysis, Validation, Investigation, Visualization, Methodology, Writing - original draft, Project administration, Writing – review and editing; Emily Dolson, Anya E Vostinar, Conceptualization, Software, Supervision, Methodology, Writing – review and editing; Luis Zaman, Conceptualization, Resources, Supervision, Methodology, Project administration, Writing – review and editing

### Author ORCIDs

Alexander Lalejini http://orcid.org/0000-0003-0994-2718
Anya E Vostinar http://orcid.org/0000-0001-7216-5283

**Decision letter and Author response**
Decision letter https://doi.org/10.7554/eLife.79665.sa1
Author response https://doi.org/10.7554/eLife.79665.sa2

## Additional files

### Supplementary files
• MDAR checklist

### Data availability
Our source code for experiments, analyses, and visualizations is publicly available on GitHub (https://github.com/amlalejini/directed-digital-evolution, copy archived at swh:1:rev:c94684a751a54bc6f2ef-2e7c0830614dc280ca01). Our GitHub repository is publicly archived using Zenodo with the following https://doi.org/10.5281/zenodo.6403135. The data produced by our computational experiments are publicly available and archived on the Open Science Framework: https://osf.io/zn63x/ (https://doi.org/10.17605/OSF.IO/ZN63X).

The following dataset was generated:

| Author(s) | Year | Dataset title | Dataset URL | Database and Identifier |
|---|---|---|---|---|
| Lalejini A | 2022 | Data from: Selection schemes from evolutionary computing show promise for directed evolution of microbes | https://osf.io/zn63x/ | Open Science Framework, 10.17605/OSF.IO/ZN63X |

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
