## [Editor Report]

The study offers a valuable contribution to the field. While the fields of artificial life and experimental evolution in microbes have been connected for many years, there have been few studies to meaningfully demonstrate how work in evolutionary computation can meaningfully inform the design and execution of microbial experiments. This study represents a truly innovative approach and may fuel further studies at the intersection between computational evolution and experimental evolution.

---

## [Decision Letter]

**Decision letter after peer review:**

Thank you for submitting your article "Artificial selection methods from evolutionary computing show promise for directed evolution of microbes" for consideration by *eLife*. Your article has been reviewed by 3 peer reviewers, and the evaluation has been overseen by a Reviewing Editor and Christian Landry as the Senior Editor. The following individual involved in review of your submission has agreed to reveal their identity: Juan Diaz-Colunga (Reviewer #2).

Essential revisions:

Please address the issues raised by reviewers 1 and 2, as they can substantially improve the manuscript. In general, the paper could benefit from more clarity in certain aspects.

*Reviewer #1 (Recommendations for the authors):*

Overall, very nice work. I have a few minor suggestions to improve the paper, as follows:

1) Please specify whether the tournament selection was with or without replacement. Additionally, please justify why you selected tournaments of size 4 (preliminary testing? Heuristic? Arbitrary?)

2) You mentioned that you chose 55,000 generations based on "the number of digital organism generations that elapsed in our directed evolution experiments". I'm confused by what you mean by this. What was the termination criterion in the directed evolution experiments that resulted in 55,000 generations? Later, you say "2,000 generations (the number of cycles in our directed evolution experiments)"; how does the 2,000 generations relate to the 55,000 generations? Please clarify. Also, since you say "digital organisms" I assume you mean in the simulated directed evolution experiments, but please insert the word "simulated" to make this clear.

3) It would be nice to have at least one graph that either summarizes or exemplifies the relative speed of evolution for the various selection methods, and some discussion of this.

4) I suspect that the relatively poor performance of NDE may be due to the large number of objectives, leading to very low selection pressure (because so many solutions are non-dominated). This is also consistent with your observation that NDE selected the most populations on average. I think some discussion of this is warranted.

*Reviewer #2 (Recommendations for the authors):*

1) Regarding my main concern on the potential presence of an unacknowledged selective pressure favoring populations of generalists, I suggest the authors study the effect of varying (or removing) the bottleneck applied when selected populations are transferred to the offspring metapopulation. In fact, I am not sure I understand why this bottleneck was even introduced in the first place. In laboratory experiments, it is common to apply these bottlenecks in serial batch culture experiments. However, the setting in this work is more similar to a turbidostat where newborn individuals replace existing ones, and thus I do not see why the parent populations could not be propagated into the offspring "as is". It would be helpful to see if a less harsh population bottleneck (or none at all) at the time of propagation would palliate the low diversity of task profiles in the three less effective protocols --- and maybe improve their performance.

2) Along the same lines, I have an additional suggestion: when the selected parent populations are propagated into the next meta-generation, repeats are allowed. Thus, two or more offspring populations from a same parent could potentially be very similar to one another (an issue that could be aggravated as more meta-generations pass). In other words, this could make it so variation is quickly exhausted as the experiment progresses. In fact, this has been observed in artificial selection experiments on microbial communities [Chang et al., Evolution 2020]. It has been suggested that perturbing the populations (for instance through the inoculation of invader species into the selected populations at the time of propagation) could replenish variation upon which selection could further act [Chang et al., Nature Ecology and Evolution 2021; Sánchez et al., Annual Review of Biophysics 2021]. For the purposes of this work, this loss of variation can simply be seen as an intrinsic limitation of the elite, top-10% and tournament treatments. But I am curious to see whether actively replenishing variation before each meta-generation could palliate the loss of diversity and improve performance. This is NOT required to support the findings of the paper as it stands, but it might serve to establish causal links between the success of a selection scheme and its ability to maintain variation in task profiles, further strengthening the results.

3) Regarding the accessibility of the paper to a broad audience: as a non-expert in evolutionary computing, some aspects of the simulations were difficult for me to follow. I think it will not be easy for readers outside of the field of evolutionary computing to build an intuition regarding how the authors' digital evolution framework could be mapped to an evolutionary process of a microbial population in a laboratory setting. I think a way to address this could be to update Figure 1 and section 3, which currently does not read well in prose. It would be very useful to expand panels b) and c) in Figure 1, which are currently a bit generic, to give the reader a better sense of what it means for the population to mature (what happens during the life span of an organism? where do inputs come from? etc.), what kind of properties of the population are assessed at the evaluation stage, and how. This should also serve to establish a clearer analogy with microbial evolutionary processes and make the manuscript more accessible to a broader audience. For the same reason, I think that describing the considered selection schemes the first time they are mentioned would be useful for non-expert readers. Currently they are described in dedicated sections, but I think that introducing them at least minimally when they are first brough up would be helpful.

---

## [Author Response]

Essential revisions:Please address the issues raised by reviewers 1 and 2, as they can substantially improve the manuscript. In general, the paper could benefit from more clarity in certain aspects.Reviewer #1 (Recommendations for the authors):Overall, very nice work. I have a few minor suggestions to improve the paper, as follows:1) Please specify whether the tournament selection was with or without replacement. Additionally, please justify why you selected tournaments of size 4 (preliminary testing? Heuristic? Arbitrary?)

We implemented tournament selection with replacement. We clarified this in our description of tournament selection in the Methods.

In our evolutionary computation work, we usually apply tournament selection in the context of larger pools of candidates for selection (e.g., populations of 1,000) than we use in our experiments here (96). Given this order-of-magnitude reduction in population size, we felt that it made sense to use an accordingly smaller tournament size. We often use a tournament size of 8 in our evolutionary computation work, and so we initially selected 4 as the next smallest power of 2. Exploratory experiments provided further support for this decision, as we found that diversity maintenance was important for evolving performant populations (just as it often is for solving problems in evolutionary computing). While standard tournament selection has no mechanism for diversity maintenance, tournament size influences the number of lineages that survive from generation-to-generation: smaller tournaments result in the selection of a larger number of distinct individuals. Though, admittedly, stable coexistence is not possible over large time scales under tournament selection regardless of tournament size [Dolson et al., 2018].

We do not think that using a different tournament size would change our overall results (relative to our controls and the multi-objective selection schemes). However, we do acknowledge that the tournament size could be tuned to improve its performance. With tuning, we would expect its performance to be no worse than either the elite or top-10% regimes for any of our experiments. We added an acknowledgement of this limitation to our conclusion.

2) You mentioned that you chose 55,000 generations based on "the number of digital organism generations that elapsed in our directed evolution experiments". I'm confused by what you mean by this. What was the termination criterion in the directed evolution experiments that resulted in 55,000 generations? Later, you say "2,000 generations (the number of cycles in our directed evolution experiments)"; how does the 2,000 generations relate to the 55,000 generations? Please clarify. Also, since you say "digital organisms" I assume you mean in the simulated directed evolution experiments, but please insert the word "simulated" to make this clear.

We absolutely see how these statements were confusing. We have clarified them as you suggest, inserting the word "simulated" where appropriate. We edited the Experimental Design subsection to further highlight the distinctions between a conventional evolutionary computing experimental setup and the setups used for our simulated directed evolution experiments. We additionally edited our mention of "2,000 generations" in the Results and Discussion to improve clarity.

We terminated each of our two simulated directed evolution experiments after 2,000 cycles of artificial selection (i.e., maturation, evaluation, selection, and propagation). During each cycle, each population of digital organisms evolves for the duration of the maturation period. During the maturation period, digital organisms produce offspring as soon as they have copied themselves and divided. As such, generations at the organism-level are asynchronous and must be measured instead of configured. We terminate the maturation period after 200 simulation updates (which corresponds to approximately 25 to 35 organism-level generations).

Our initial control experiment, however, is set up as a conventional evolutionary computing system. There is no metapopulation, and "organism-level" generations are synchronous. We terminated this conventional evolutionary computation experiment after 55,000 generations of evolution (i.e., program evaluation, selection, and reproduction). We chose to terminate this experiment after 55,000 generations to approximate the total depth of evolutionary search that occurred in our simulated directed evolution experiments (i.e., approximately 55,000 organism-level generations). Later on, in the Results and Discussion, we state that just 2,000 generations of our conventional evolutionary computing experiment would have been sufficient to evolve more performant programs than those that evolved in our simulated directed evolution experiment.

3) It would be nice to have at least one graph that either summarizes or exemplifies the relative speed of evolution for the various selection methods, and some discussion of this.

We agree, and we appreciate this suggestion. We added a time series for both single-population and metapopulation task coverage for each of our experiments (Figure 2, 3, and 4), and we included some discussion of these data in our Results and Discussion section as well as in the Conclusion. These data highlight how rapidly directed microbial evolution experiments might benefit from using lexicase or a Pareto-based selection method, which is especially promising given the amount of effort required to carry out a single round of artificial selection in the laboratory. We would not have thought to include this without your suggestion!

Additionally, we switched to a more accessible color scheme for our figures, as the time series graphs require readers to distinguish between conditions via color.

4) I suspect that the relatively poor performance of NDE may be due to the large number of objectives, leading to very low selection pressure (because so many solutions are non-dominated). This is also consistent with your observation that NDE selected the most populations on average. I think some discussion of this is warranted.

Absolutely, we completely agree. We added a paragraph of discussion to the

"Lexicase and non-dominated elite selection show promise for directed evolution" subsection of the Results and Discussion, including references to relevant work.

Reviewer #2 (Recommendations for the authors):1) Regarding my main concern on the potential presence of an unacknowledged selective pressure favoring populations of generalists, I suggest the authors study the effect of varying (or removing) the bottleneck applied when selected populations are transferred to the offspring metapopulation. In fact, I am not sure I understand why this bottleneck was even introduced in the first place. In laboratory experiments, it is common to apply these bottlenecks in serial batch culture experiments. However, the setting in this work is more similar to a turbidostat where newborn individuals replace existing ones, and thus I do not see why the parent populations could not be propagated into the offspring "as is". It would be helpful to see if a less harsh population bottleneck (or none at all) at the time of propagation would palliate the low diversity of task profiles in the three less effective protocols --- and maybe improve their performance.

We fully agree that the effects of bottleneck size are an important direction of further study, especially as we expand our model to incorporate more complex interactions between digital organisms. We configured a bottleneck size of 1% for two reasons. The first is as you state: bottlenecks are common in serial batch culture experiments, which we are using for our initial proof of concept laboratory experiments.

Second, in early exploratory experiments, we tried several "sampling sizes" for propagating selected populations. As you suggest, we fully expected that propating entire populations would be best, as we would prevent the loss of potentially important diversity. To our great surprise, depending on the selection method, sample size either had minimal effect or the 1% sample size outperformed larger sample sizes (see Author response image 1). Admittedly, we conducted these exploratory experiments well before the final set of experiments presented in our manuscript, so their setups are not the same: we only compared elite, lexicase, non-dominated elite, and a no selection control; the environment is simpler with 8 population-level functions instead of 18; the maximum population size is 900 instead of 1,000; the maturation period is longer (300 updates versus 200); and we ran the experiment for fewer cycles (500 instead of 2,000).

There is perhaps some precedent for our observations about sample size in recent evolution experiments using populations of *E. coli* where [Izutsu et al., 2021] observed more rapid and greater fitness gains in populations subjected to 100- and 1000-fold dilution regimes than populations subjected to an 8-fold dilution regime. As we expand our model to include more interactions, we expect our choice of sample size to matter more, as smaller samples may fail to propagate all members of an important interaction required for community function.

To address this recommendation, we expanded our supplemental material to include the results for our exploratory experiment where we varied the propagation sample size. We reference these data as a justification for our choice of sample size in the Digital Directed Evolution section. In the Conclusion, we added some discussion of the value of future work that explores different laboratory setups/constraints (e.g., bottleneck size, maturation periods, metapopulation size).

**Author response image 1. sa2fig1:** 

2) Along the same lines, I have an additional suggestion: when the selected parent populations are propagated into the next meta-generation, repeats are allowed. Thus, two or more offspring populations from a same parent could potentially be very similar to one another (an issue that could be aggravated as more meta-generations pass). In other words, this could make it so variation is quickly exhausted as the experiment progresses. In fact, this has been observed in artificial selection experiments on microbial communities [Chang et al., Evolution 2020]. It has been suggested that perturbing the populations (for instance through the inoculation of invader species into the selected populations at the time of propagation) could replenish variation upon which selection could further act [Chang et al., Nature Ecology and Evolution 2021; Sánchez et al., Annual Review of Biophysics 2021]. For the purposes of this work, this loss of variation can simply be seen as an intrinsic limitation of the elite, top-10% and tournament treatments. But I am curious to see whether actively replenishing variation before each meta-generation could palliate the loss of diversity and improve performance. This is NOT required to support the findings of the paper as it stands, but it might serve to establish causal links between the success of a selection scheme and its ability to maintain variation in task profiles, further strengthening the results.

Yes, we fully agree with you that the loss of variation is an intrinsic limitation of the elite, top-10%, and tournament selection methods. This is a fantastic observation because it beautifully connects a large body of evolutionary computing research with recent laboratory experiments in [Chang et al., Evolution 2020]. Thank you for this reference! In fact, your observation is one of the original motivations for our study: the artificial selection protocols that we found to be commonly used in directed microbial evolution have no algorithmic mechanisms for maintaining diversity.

The rapid loss of variation intrinsic to elite, top-10%, and tournament selection is well-established in evolutionary computing as a reason for "premature convergence", which essentially means exactly what happens in [Chang et al., Evolution 2020]: adaptive evolution stagnates because the population converges to a small region of the fitness landscape (as a direct result of these "elitist" selection schemes failing to maintain meaningful diversity).

We appreciate the pointer to related work on replenishing variation by perturbing populations. There are strong parallels between directed microbial evolution and evolutionary computing on this front. Similar approaches (though not identical) have been developed in evolutionary computing for exactly the same goal of replenishing variation to avoid "adaptive stagnation". E.g., injecting randomly generated individuals into the population each generation, migrating individuals from different subpopulations at regular intervals.

We are also very much curious to see whether actively replenishing variation before each generation would have disproportionate effects on any particular selection protocol. It is not obvious to us whether replenishing variation would improve performance in the elite, top-10%, or tournament selection regimes more so than it would in the non-dominated elite and lexicase regimes. Which is all to say, it would make for an interesting experiment! At least in evolutionary computation, the *kinds* of perturbations (or introduced variation) often matter as much as whether or not they are included in the first place. We would not be surprised if this is also true for microbial systems.

To address this recommendation in our manuscript, we explicitly call out this intrinsic limitation of elite, top-10%, and tournament selection the Results and Discussion, including a reference to similar findings in [Chang et al., Evolution 2020]. We incorporated your suggestion to explore population perturbations as a method to replenish variation as a future direction in the Conclusion.

3) Regarding the accessibility of the paper to a broad audience: as a non-expert in evolutionary computing, some aspects of the simulations were difficult for me to follow. I think it will not be easy for readers outside of the field of evolutionary computing to build an intuition regarding how the authors' digital evolution framework could be mapped to an evolutionary process of a microbial population in a laboratory setting. I think a way to address this could be to update Figure 1 and section 3, which currently does not read well in prose. It would be very useful to expand panels b) and c) in Figure 1, which are currently a bit generic, to give the reader a better sense of what it means for the population to mature (what happens during the life span of an organism? where do inputs come from? etc.), what kind of properties of the population are assessed at the evaluation stage, and how. This should also serve to establish a clearer analogy with microbial evolutionary processes and make the manuscript more accessible to a broader audience. For the same reason, I think that describing the considered selection schemes the first time they are mentioned would be useful for non-expert readers. Currently they are described in dedicated sections, but I think that introducing them at least minimally when they are first brough up would be helpful.

We agree. We have updated Figure 1 to make it a little bit less generic and hopefully more intuitive. We also edited the description of our model of directed evolution, including our description of digital organisms ("Digital Organisms" subsection of Digital Directed Evolution). In general, we tried to provide extra intuition for our model by highlighting potential analogues in microbial evolution systems. As suggested, we gave a brief overview of the evolutionary computing selection schemes where they are first mentioned in the introduction.

References

Chang, C.-Y. et al. Engineering complex communities by directed evolution. Nat Ecol Evol 5, 1011–1023 (2021).

Chang, C., Osborne, M. L., Bajic, D. and Sanchez, A. Artificially selecting bacterial communities using propagule strategies. Evolution 74, 2392–2403 (2020).

Dolson, E. L., Banzhaf, W. and Ofria, C. Ecological theory provides insights about evolutionary computation. https://peerj.com/preprints/27315 (2018) doi:10.7287/peerj.preprints.27315v1.

Izutsu, M., Lake, D. M., Matson, Z. W. D., Dodson, J. P. and Lenski, R. E. Effects of periodic bottlenecks on the dynamics of adaptive evolution in microbial populations.